# Craniodental divergence associated with bite force between hybridizing pine squirrels (*Tamiasciurus*)

Dylan M. Poorboy[1]*, Jonathan J.-M. Calède[1,2], Andreas S. Chavez[1,3]

**1** Department of Evolution, Ecology and Organismal Biology, The Ohio State University, Columbus, OH, United States of America, **2** The Ohio State University at Marion, Marion, OH, United States of America, **3** Translational Data Analytics Institute, The Ohio State University, Columbus, OH, United States of America

* poorboy.1@buckeyemail.osu.edu

**Data Availability Statement:** Our data have now been made accessible through Dryad. The digital object identifier for our dataset is as follows: doi:10.5061/dryad.kd51c5bbg.

## Abstract

Bite force can be a limiting factor in foraging and can significantly affect the competitive ability and lifetime fitness of mammals. *Tamiasciurus* squirrels feed primarily on conifer seeds and have a strong bite force to mechanically extract seeds from conifer cones with their mouths. In the North Cascades region, Douglas squirrels (*Tamiasciurus douglasii*) and red squirrels (*T. hudsonicus*) occupy ecologically different forests with different hardnesses in conifer cones. The ranges of these species overlap in a narrow hybrid zone where these forests meet near the crest of the North Cascades. We examined interspecific divergence in dietary ecomorphology in allopatry, in sympatry within the hybrid zone, and between hybrids and each parental species. We focused on three craniodental traits, including the incisor-strength index as a proxy measure for maximal bite force, cranial-suture complexity, and mandible shape. We find that these sister squirrel species differ in bite force and suture complexity in allopatry and sympatry and that mandible shape changes with the expected hardness of accessed food items, but is not significantly different between species. Furthermore, we find that hybrids display morphologies that overlap with hybrid zone red squirrels but not with hybrid zone Douglas squirrels. This work shows how important ecological processes at shallow evolutionary timescales can impact the divergence of morphological traits in taxa with extreme conservation of craniomandibular shape.

## Introduction

The ecological processes that generate morphological divergence can be challenging to discern in comparative studies focused on species across deep taxonomic timescales. Ecomorphology can be the product of several, at times opposing, selective processes that may not persist through the entire evolutionary history of deep lineages. At shallow timescales, different processes, including divergent ecological selection of morphology and behavior, hybridization, and interspecific competition, can be important initiators of ecomorphological diversity [1–3]. Hybrid zones representing secondary contact and hybridization between ecologically divergent species are practical arenas to investigate how these three different processes impact

**Funding:** The authors received no specific funding for this work.

**Competing interests:** The authors have declared that no competing interests exist.

variation on ecomorphology at shallow timescales. In hybrid zones that span ecological transitions, differences in phenotypic variation between species is subject to the homogenizing effects of gene flow and its counterposing force, divergent selection [4–6]. Furthermore, interspecific competition within hybrid zones can potentially drive the divergent evolution through character displacement [7–9] or, in contrast, phenotypic convergence driven by shared ecological pressures [10–12].

Hybridization, in the absence of evolutionary forces, can have significant consequences on phenotypic variation. Hybridization leads to novel epistatic interactions between differentiated genomes, which may allow for a change in morphospace occupation due to a relaxation of trait covariation [13, 14]. Thus, hybridization can immediately lead to (1) hybrid phenotypes that are intermediate between the parental phenotypes [15, 16]; (2) hybrid phenotypes that resemble one of the parental phenotypes, reflecting genetic dominance [17–19]; or (3) transgressive hybrid phenotypes that are more extreme than either parental phenotype [20–24]. The consequences of evolutionary forces on hybrid phenotypes are also varied and range from strong selection against hybrid morphologies [25] to selection that favors novel hybrid morphologies [2, 3].

Ecomorphological traits associated with foraging are routinely under strong natural selection for functional performance and illustrate the combination of both precise local adaptation and extreme evolutionary diversification [26–29]. In mammals, several craniodental traits are associated with generating and sustaining maximal bite force. First, the incisor strength index is a reliable proxy for maximal bite force in rodents [30]. This index is based on a rectangular cross-section of the incisor and its ability to resist a bending force, which allows it to be a valuable estimator of bite force where *in vivo* measurements are challenging to acquire or consistently replicate due to variable behavior [30]. This index was developed from interspecific studies in deeper evolutionary time than is addressed herein; however, we expect it to be functional as a proxy measure of comparing relative biting strength, although it may not be directly translatable to units of force. Next, the suture lines of the cranium reveal the counterforce stresses that the cranium receives from biting on food items [31–35]. Consequently, suture complexity is positively associated with a greater ability to absorb and redistribute the shocks that result from stronger bite force [36, 37]. Finally, mandible shape is another demonstrated predictor of bite force [38] and may influence masticatory muscle attachment and mechanical levers, with the expectation that more durophagous animals will have larger or more robust mandibular processes and higher mechanical lever advantages at the incisors [28, 39–41].

The hybrid zone between ecologically divergent Douglas squirrels (*Tamiasciurus douglasii*) and red squirrels (*T. hudsonicus*) represents an exemplary opportunity to study the consequences of ecologically divergent selection, interspecific competition, and hybridization on craniodental traits associated with bite force. This parapatric pair of squirrel species diverged less than a million years ago and are hybridizing in a narrow secondary contact zone near the crest of the North Cascades region in northern Washington and southern British Columbia (Fig 1; [42–44]). *Tamiasciurus* squirrels are strongly associated with coniferous forests, feed primarily on conifer seeds by mechanically removing cone scales with their mouth, are mostly asocial, and will vigorously defend territories to protect their centralized cache of conifer cones [42, 45].

In the greater region outside of the hybrid zone, major habitat differences between Douglas squirrels and red squirrels implicate ecological divergent selection as an important driver of divergence in craniodental traits associated with bite force. Douglas squirrels occupy wet maritime forests on the west side of the Cascade Range, and red squirrels occupy dry continental forests on the east side. These habitat differences correspond with differences in the hardness of conifer cones. In the dry continental forests on the east side, lodgepole pine seeds are an

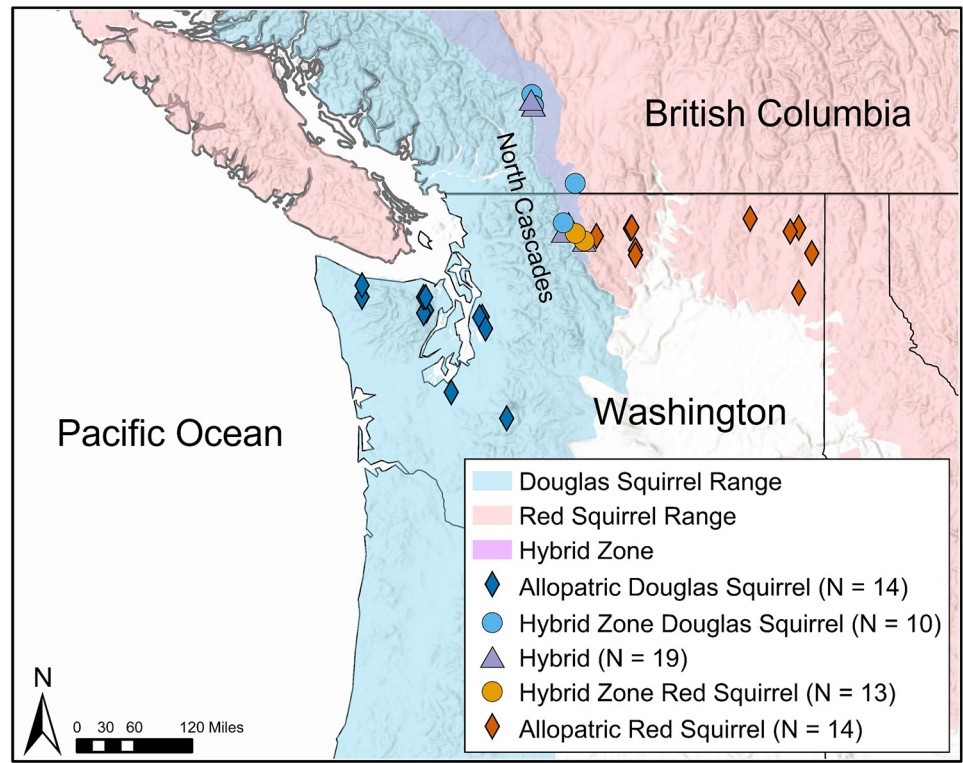

**Fig 1. Map indicating approximate range boundaries for Douglas squirrels (blue), red squirrels (red), and the hybrid zone (purple) in the North Cascades region.** Range shapefiles developed from published descriptions [49, 50] by A.S. Chavez. Symbols indicate sampling localities. Some specimens within the hybrid zone geographically overlap, see S1 Table for complete specimen details. Basemap reprinted from Terrain With Labels under a CC BY license, with permission from ESRI, original copyright 2016.

important food for red squirrels and are encased in hard, serotinous (closed) cones. In contrast, all conifer species in the wet maritime forests in the range of Douglas squirrels produce qualitatively softer and more opened cones [46–48]. One of the most striking craniodental differences between these squirrel species that may be associated with their different biomechanical challenges of procuring conifer seeds is the prominent sagittal crest found in red squirrels and the shorter and medio-laterally expanded sagittal "plateau" found in Douglas squirrels [43, 45]. This difference in sagittal crest development is expected to drive a difference in produced bite force for these species, as it relates to muscle attachment space for the temporalis muscle involved in the biting motion. Comparisons of other craniodental traits associated with bite force between these squirrel species living in different forests will help inform whether or not divergent selection has generated greater divergence in other parts of the skull than previously understood.

Within the hybrid zone, observational studies have shown that both parental squirrel species and their hybrids are syntopic and select similar food items in subalpine forests dominated by Engelmann spruce and subalpine fir [45, 51], which are expected to be qualitatively soft cones. Hybrids can also backcross successfully with both parental species [43]. The territorial behavior and overlapping niche of both parental species and their hybrids within the hybrid zone indicate that interspecific competition among all squirrel types is potentially strong, which may have important consequences on access to food resources and, ultimately, the evolution of craniodental morphologies associated with foraging. One potential outcome of

intense interspecific competition is the evolution of ecological character displacement in response to selection that lessens resource competition between interacting species and leads to specialization towards different resources [7]. However, character displacement and competitive exclusion in hybrid-zone systems may be prevented by the co-existence and viability of hybrids with intermediate phenotypes [52]. Alternatively, these species may converge in their ecomorphologies due to the selective pressures of a shared food resource. Comparisons of ecomorphologies between hybrid zone populations and allopatric populations for each parental squirrel species and their hybrids can begin to uncover whether interspecific competition or convergent evolution are important factors driving morphological change. Furthermore, comparisons of ecomorphologies between hybrids and both parental species can help reveal how the creative effects of hybridization may lead to ecomorphologies outside of or within the phenotypic range of the parental species.

Here, we conduct a comparative study of bite force strength, suture complexity, and mandible shape to determine whether divergence in these functional traits is consistent with ecological divergent selection, interspecific competition, and hybridization. Specifically, we investigate the following three hypotheses on the patterns of ecomorphological differences associated with bite force: (1) differences between allopatric Douglas squirrels and red squirrels are consistent with divergent selection due to major differences in cone hardness; (2) ecomorphologies between Douglas squirrels and red squirrels in sympatry within the hybrid zone are convergent, consistent with a shared ecology; and (3) hybrid morphologies are overlapping with or intermediate to either parental species due to genetic admixture between the parental species and a shared ecology.

## Results

### Bite force

The bite force quotient (BFQ) of the lower incisor differs significantly across squirrel groups (Fig 2; $p < 0.0001$, $F_{4,55} = 8.86$). Pairwise comparisons reveal that the BFQ is lower for Douglas squirrels than for red squirrels, both in allopatry ($p < 0.0001$) and in sympatry within the hybrid zone ($p = 0.014$). The hybrid zone red squirrel population has a lower BFQ than the allopatric red squirrel population ($p = 0.014$). Similarly, the hybrid zone Douglas squirrel population has a lower BFQ than the allopatric Douglas squirrel population ($p = 0.036$). The BFQ of hybrids is not significantly different from that of sympatric hybrid-zone red squirrels ($p = 0.94$), but it is significantly greater than that of sympatric hybrid-zone Douglas squirrels ($p = 0.016$).

### Suture morphology

We find significant differences in suture complexity (LR) among squirrel groups for the premaxillofrontal (Fig 3B; $p < 0.001$, $F_{4,54} = 6.81$), maxillofrontal (Fig 3D; $p = 0.0027$, $F_{4,53} = 4.64$), and sagittal sutures (Fig 3F; $p < 0.0001$, $F_{4,50} = 10.20$), but no differences for either the nasofrontal (Fig 3C; $p = 0.182$, $F_{4,54} = 1.62$) or coronal sutures (Fig 3E; $p = 0.223$, $F_{4,53} = 1.48$). Our post-hoc pairwise comparisons demonstrate significant differences between allopatric red squirrels and allopatric Douglas squirrels for the premaxillofrontal ($p = 0.0041$), maxillofrontal ($p = 0.0021$), and sagittal ($p = 0.0002$) sutures. We also find significant pairwise differences between the hybrid-zone red squirrels and the hybrid-zone Douglas squirrels for the premaxillofrontal ($p = 0.0013$), maxillofrontal ($p = 0.037$), and sagittal ($p = 0.01$) sutures. Hybrids are only significantly different from the hybrid-zone Douglas squirrels in their premaxillofrontal ($p = 0.0047$) and sagittal sutures ($p = 0.028$), but not in their maxillofrontal suture ($p = 0.36$). Hybrids are not significantly different from hybrid-zone red squirrels in any of their sutures.

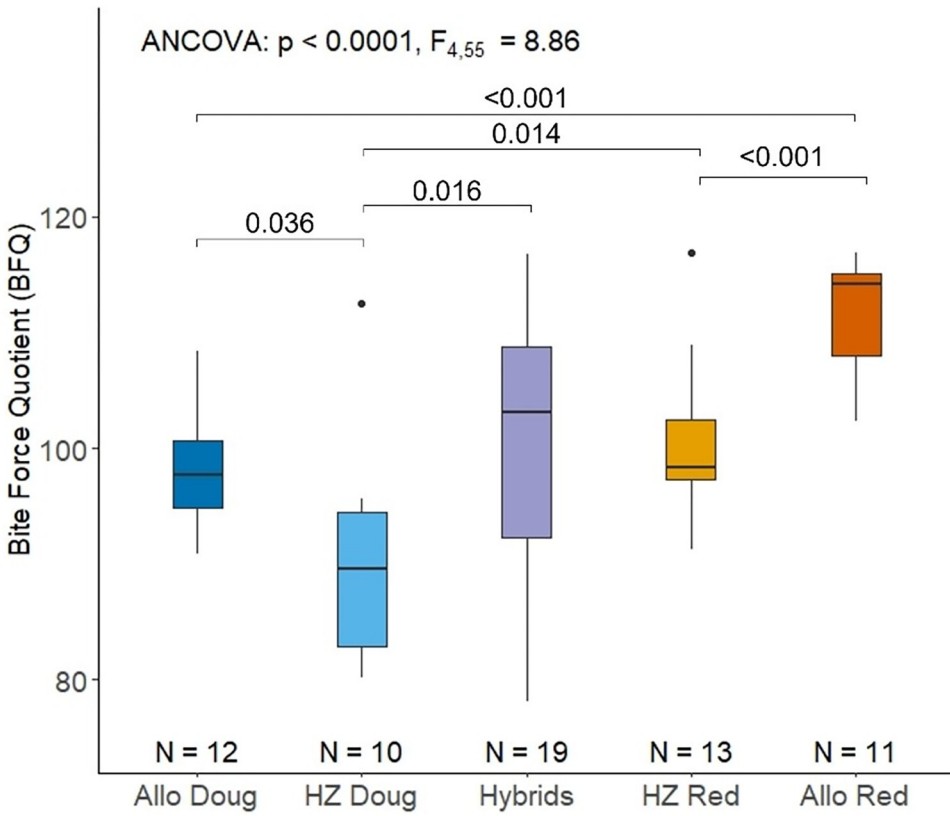

**Fig 2. Boxplot showing variation in bite force quotient (BFQ) for each group of squirrels.** Values on comparison bars represent significant p-values from pairwise comparisons. Outliers are indicated by black dots. Abbreviations: Allopatric Douglas squirrel, Allo Doug; Hybrid Zone Douglas squirrel, HZ Doug; Hybrid Zone red squirrel, HZ Red; and allopatric red squirrel, Allo Red.

The first two axes of the PCA of all five sutures, PC1 and PC2, explain 65.95% of the variation in suture complexity (Fig 3A; PC1: 45.95%; PC2: 20.00%). PC1 is primarily driven by the complexity of the premaxillofrontal, maxillofrontal, and sagittal sutures with negative PC scores representing greater complexity for these three sutures. PC2 is primarily determined by variation in coronal and nasofrontal sutures; with increased complexity in the nasofrontal suture associated with positive PC scores, whereas increased complexity in coronal sutures is associated with negative PC scores. An ANCOVA reveals a significant difference in PC1 scores among squirrel groups (Group: $p < 0.0001$, $F_{4,45} = 13.38$; sex: $p = 0.002$, $F_{1,45} = 10.47$; size: $p = 0.42$, $F_{1,45} = 0.67$), but no significant difference is present for PC2 among groups (Group $p = 0.528$, $F_{4,45} = 0.81$; sex: $p = 0.97$, $F_{1,45} = 0.002$; size: $p = 0.38$, $F_{1,45} = 0.80$). Post-hoc tests reveal a significant difference in PC1 scores between Douglas squirrels and red squirrels in allopatry ($p = 0.0002$), as well as in sympatry ($p = 0.013$), however, no significant differences in PC1 scores between allopatric Douglas squirrel and hybrid-zone Douglas squirrels ($p = 0.73$) or between allopatric red squirrels and hybrid-zone red squirrels ($p = 0.84$). Hybrids differ significantly in PC1 scores from hybrid-zone Douglas squirrels ($p = 0.0379$), but not from hybrid-zone red squirrels ($p = 0.386$).

## Geometric morphometrics

Monte Carlo randomization supported PC1 ($p < 0.001$) and PC2 ($p < 0.001$) as significant axes for the morphospace PCA, with all other axes being nonsignificant ($p > 0.99$); only the

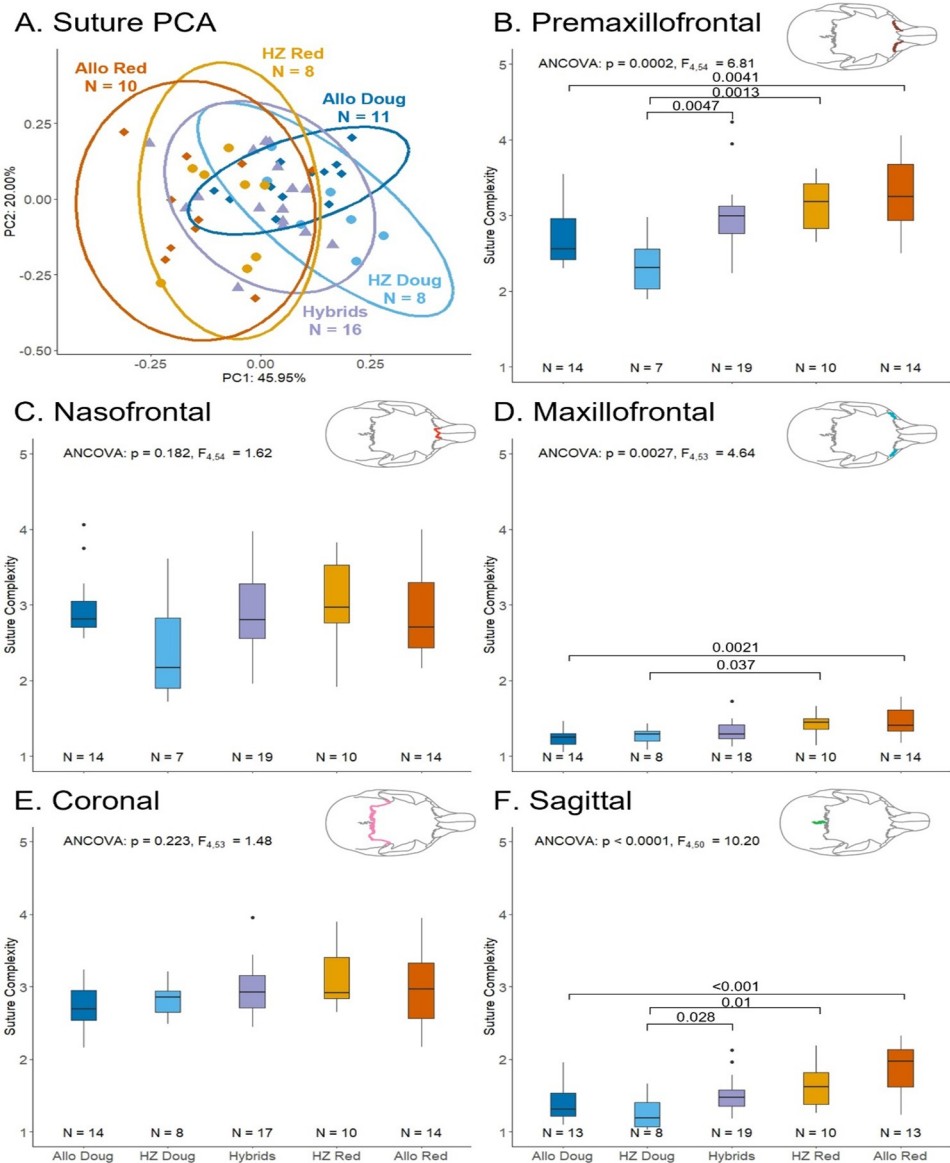

**Fig 3. Suture complexity comparisons between *Tamiasciurus* groups.** Values on comparison bars represent significant p-values from pairwise comparisons. Outliers are indicated by black dots. Abbreviations are provided in Fig 2. (A) Principal component analysis for all five sutures. (B) ANCOVA for the premaxillofrontal suture. (C) ANCOVA for the nasofrontal suture. (D) ANCOVA for the maxillofrontal suture. (E) ANCOVA for the coronal suture. (F) ANCOVA for the sagittal suture.

first two axes were retained for subsequent analyses. PC1 and PC2 explain 45.97% of the variation in mandible morphology among the squirrels studied (Fig 4; PC1: 35.57%; PC2: 10.40%). Specimens with more positive PC1 values have an anteroposteriorly shortened mandible, a dorsoventrally shortened condylar process, an dorsoventrally heightened enlarged angular process, and a higher coronoid process. Positive PC2 values are associated with a short ramus, including a reduced coronoid process and a short angular process.

We analyzed the possible effects of centroid size and sex on mandible shape to ascertain the roles of allometric scaling and sexual dimorphism. The Procrustes ANOVA of PC1 and PC2 showed that mandible shape is significantly correlated with log10-transformed centroid size

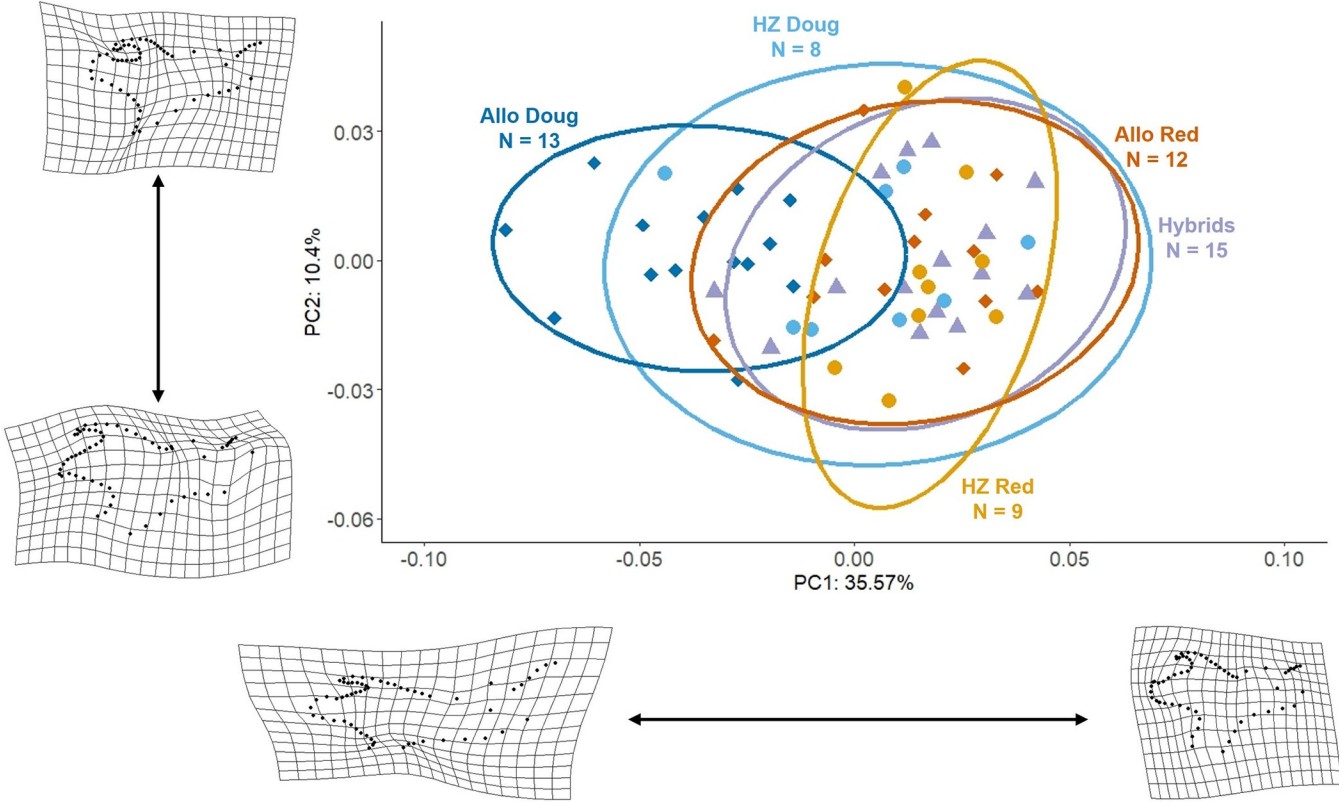

**Fig 4. Principal component analysis of mandible shape in lateral view.** Deformation grids for principal component axes accompany their respective axis and were based on the graphed extremes for the figure. Abbreviations are provided in Fig 2.

(Table 1; p = 0.001, $F_{1,46}$ = 5.19) and the interaction between group and centroid size (p = 0.007, $F_{4,46}$ = 2.38); but mandible shape was not significantly affected by sex (p = 0.264, $F_{1,46}$ = 0.61) or interactions between sex and size (p = 0.166, $F_{1,46}$ = 0.99). The ANOVA of centroid size demonstrated a significant difference in size among the different squirrel groups (Fig 5; p < 0.001, $F_{4,53}$ = 22.94). Post-hoc comparisons indicate significant differences in centroid size between allopatric red squirrels and allopatric Douglas squirrels (p < 0.0001), as well as between hybrid-zone red squirrels and hybrid-zone Douglas squirrels (p = 0.0011). In addition, hybrid-zone Douglas squirrels are significantly larger than allopatric Douglas squirrels (p<0.0001). There are no significant differences between hybrid-zone red squirrels and allopatric red squirrels (p = 0.498). Hybrids are significantly larger than hybrid-zone Douglas

**Table 1. Procrustes ANOVA of lateral view mandible shape data.** Relationships assessed were for squirrel group, sex, log transformed centroid size, and the interactions size by group and size by sex. Significant p-values are shown in bold.

|  | Df | SS | MS | Rsq | F | Z | Pr(>F) |
|---|---|---|---|---|---|---|---|
| Group | 4 | 0.002426 | 0.000606 | 0.38768 | 13.3115 | 6.4123 | **0.001** |
| Sex | 1 | 5.74E-05 | 5.74E-05 | 0.00918 | 1.2603 | 0.6071 | 0.264 |
| Size | 1 | 0.001139 | 0.001139 | 0.18198 | 24.9946 | 5.1933 | **0.001** |
| Group:Size | 4 | 0.000455 | 0.000114 | 0.07271 | 2.4967 | 2.3804 | **0.007** |
| Sex:Size | 1 | 8.47E-05 | 8.47E-05 | 0.01353 | 1.8587 | 0.9875 | 0.166 |
| Residuals | 46 | 0.002095 | 4.56E-05 | 0.33492 |  |  |  |
| Total | 57 | 0.006257 |  |  |  |  |  |

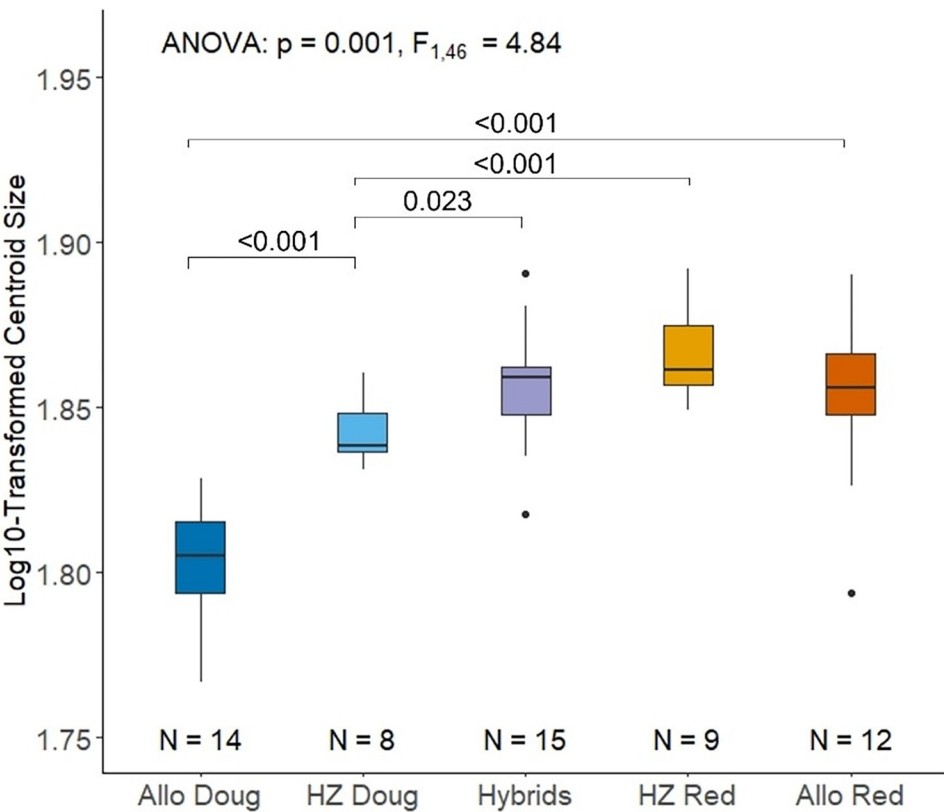

**Fig 5. Variation in log10-transformed mandible centroid size across squirrel groups.** Values on comparison bars represent significant p-values from pairwise comparisons. Outliers are indicated by black dots. Abbreviations are provided in Fig 2.

squirrels (p = 0.022) but are not different from hybrid-zone red squirrels (p = 0.13). The Procrustes ANOVA indicates a significant effect of group on mandible shape (p = 0.001, $F_{4,46}$ = 6.41); however, there were no significant pairwise differences associated with Procrustes data after correction for multiple testing.

The analysis of morphospace overlap using nicheROVER reveals varying degrees of overlap in mandible shape and size among squirrel groups (Fig 6; S1 Fig). Allopatric Douglas squirrels show a high degree of overlap with hybrid-zone Douglas squirrels but low overlap with allopatric red squirrels, hybrid-zone red squirrels, and hybrids. Hybrid-zone Douglas squirrels show a high degree of overlap with hybrids and a greater probability of overlap with red squirrel populations than with Douglas squirrels in the parental range. Hybrids have a strong probability of overlapping with all other squirrel groups except for allopatric Douglas squirrels. Finally, both allopatric red squirrels and hybrid-zone red squirrels have a high overlap probability with other squirrel groups, except allopatric Douglas squirrels.

## Hybrid admixture regressions

A minimum admixture score of 0.18 and maximum admixture score of 0.83 were calculated for the hybrids studied, with a mean score of 0.5353 (± 0.0749 95% CI) for the 19 hybrid individuals. Regressions of admixture proportion for hybrids and their measured dietary ecomorphological traits were tested for genetic correlations. Admixture proportions (Q) were not significantly correlated with BFQ ($R^2$ = 0.01, p = 0.771) or any of the five LR measurements

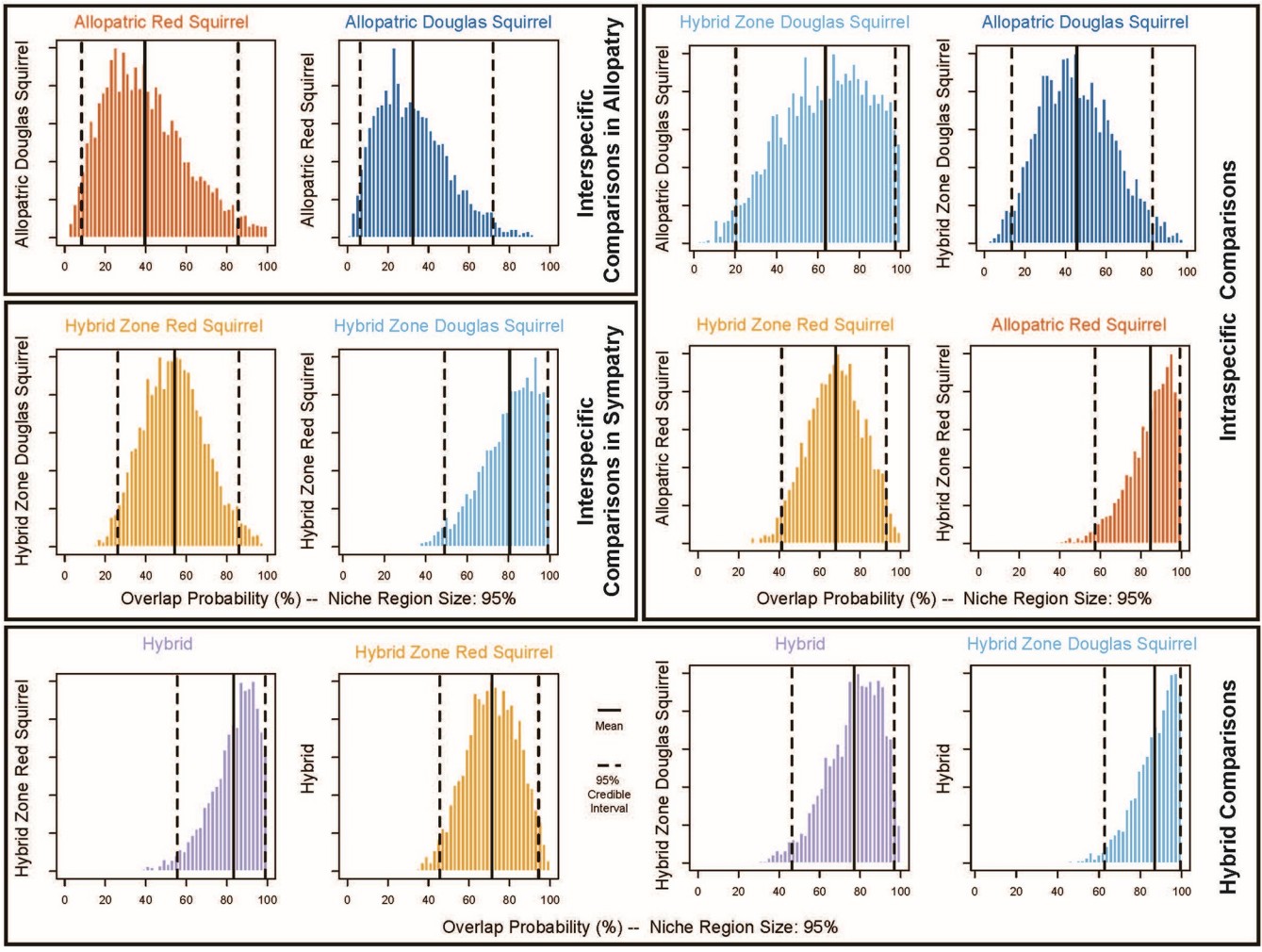

**Fig 6. The probability of mandible morphospace overlap between squirrel groups for geometric morphometric principal components 1 and 2 representing shape and log10-transformed centroid size representing size.** Data shown are a posterior histogram of 3000 iterations of overlap metric calculations, whereby the x-axis of each pairwise comparison is the range of overlap probabilities in percentage, and the y-axis is the frequency of that probability having been observed in the calculation. Mean overlap probability is shown as a solid black line, and 95% confidence intervals are shown as dashed lines for all pairs. The plots shown here represent key comparisons for this study; additional comparisons are shown in S1 Fig.

(nasofrontal: $R^2$ = 0.01, p = 0.627; premaxillofrontal: $R^2$ = 0.001, p = 0.879; maxillofrontal: $R^2$ = 0.02, p = 0.558; coronal: $R^2$ = 0.04, p = 0.464; sagittal: $R^2$ = 0.03, p = 0.468). Admixture was also not significantly correlated with PC1 of the morphospace PCA ($R^2$ = 0.08, p = 0.314) or centroid size ($R^2$ = 0.004, p = 0.811).

## Discussion

The dietary ecomorphology of an organism is phenotypically complex, involving multiple integrated morphological traits that enable interactions with the environment [53–57]. This is especially relevant for mammal skulls, where food acquisition and processing are determined mainly by muscle attachments, muscle masses, and mechanical levers that control bite force and, therefore, the food items that can be accessed [58–60]. In this study, we investigated the evolution of three morphological traits associated with feeding, including incisor shape as a measure of bite force, cranial suture complexity, and mandible shape. We found a consistent

pattern across all morphological traits that red squirrels have traits for stronger BFQ than Douglas squirrels in both allopatry and sympatry, as well as a tendency for the hybrids to more closely resemble red squirrels than Douglas squirrels in morphology.

### Inter-specific ecomorphological differences in allopatry

We find support for our hypothesis that ecomorphological divergence between allopatric red squirrels and Douglas squirrels is consistent with expectations under strong ecological divergent selection; this is true for all morphological traits studied. Red squirrels possess a stronger bite force (as measured by their incisor strength index), as well as greater suture complexity than Douglas squirrels. In these allopatric populations, it has been previously documented that red squirrels possess a prominent sagittal crest, associated with an expectedly larger temporalis muscle for stronger gnawing of hard conifer cones, whereas Douglas squirrels display a lower sagittal "plateau" [43, 45]. The sagittal crest forms along the sagittal suture. Thus, our findings of greater sagittal suture complexity in red squirrels are consistent with their stronger temporalis muscles exerting greater forces to this suture. The relationship between greater biting forces, the sagittal suture, and sagittal crest size could be addressed in future studies accompanied by temporal muscle dissections to quantify muscle attachment space. Furthermore, we also find greater complexity in the premaxillofrontal and maxillofrontal sutures in red squirrels. Greater cranial suture complexity is expected to be important in alleviating the forces associated with a strong biting action [31–35]. These specific sutures are associated with tooth-bearing bones (premaxilla and maxilla) and are specifically expected to be more complex in response to increased masticatory strains [61]. The stronger biting ability of red squirrels is also apparent in their mandible shape. Allopatric red squirrels have a shorter mandibular corpus, a dorsoventrally shorter condylar process, a dorsoventrally longer angular process, and a higher coronoid process than allopatric Douglas squirrels. This morphological shape in red squirrels enables a relatively greater mechanical advantage. This pattern is consistent with the general pattern defining nut-eating squirrels in which they have been characterized as having a robust mandibular corpus, high ramus, and well-developed angular and coronoid processes [39]. Thus, the foundation of morphological divergence between allopatric red squirrels and Douglas squirrels appears to be occurring in the components of the mandible most directly associated with a durophagous diet. Our findings of ecomorphological divergence in the craniomandibular shape between *Tamiasciurus* species are interesting in the context of skull evolution in squirrels, which have skulls that are generally considered to be extreme examples of morphological conservation [28, 39, 62–65].

The morphological differences in red squirrels that enable stronger bite force is consistent with the greater hardness of their primary food item. Seeds from lodgepole pine are a primary food item for red squirrels east of the Cascade Range. The dry climate in this region allows for high-intensity fires that threaten forest establishment. As a consequence, lodgepole pine has evolved the ability to produce serotinous (closed) cones that only open following forest fires to protect their seeds during fires. This trait is variable albeit important for lodgepole pine throughout the Rocky Mountain region, including the eastern Cascade Range (pers. comm.). Serotinous cones are harder in texture and have reduced seed numbers than non-serotinous cones [66–68]. However, the hard lodgepole pinecone creates an important selective pressure on red squirrels to evolve greater biting strength to access the seeds from the cones [46]. In contrast, Douglas squirrels live in the mesic coastal forests west of the Cascade Range, where lodgepole pine is rare or nonexistent and where serotiny is not a seed preservation strategy for any conifer species [42]. Thus, Douglas squirrels feed largely on seeds from relatively soft-

coned conifer species, such as Douglas fir *(Pseudotsuga menziesii)*, that do not require biting strengths that are as demanding.

Our study reveals an interesting difference in morphological variation for both suture complexity and mandible shape between these *Tamiasciurus* species. Red squirrels show greater variation in both morphospace and suture complexity than Douglas squirrels, indicating their greater range of phenotypic variation in dietary ecomorphology. Morphology can become more variable along an adaptive plateau rather than a sharp adaptive peak, contributing to such signals of greater variation [65, 69]. The greater range of morphological variation in red squirrels and hybrids than in Douglas squirrels may reflect a wider adaptive plateau and a greater range of viable phenotypes for suture complexity and mandible shape.

## Inter-specific ecomorphological differences in sympatry within the hybrid zone

Where red squirrels and Douglas squirrels are sympatric within the hybrid zone, we find eco-morphological divergence patterns similar to the patterns observed in allopatry outside the hybrid zone. Indeed, despite competing for the same food resources and experiencing the same dietary biomechanical needs within the hybrid zone, red squirrels maintain greater bite force and suture complexity than Douglas squirrels. The persistence of morphological differences in sympatry suggests that divergence in the genetic architecture of some craniomandibular traits is maintained. This occurs despite potential homogenization effects of backcrossed hybridization, introgression of genetic variation, and perhaps some limitations in the role of phenotypic plasticity allowing convergent morphologies in a shared environment. Also of note is the decrease in bite force for both species within the hybrid zone when compared with their conspecifics outside of the hybrid zone. This may suggest weaker biomechanical demands presented by conifer cones within the hybrid zone. This is potentially due to the hybrid zone containing both large tracts of subalpine forests that are dominated by the soft-coned Engelmann spruce *(Picea engelmannii)*, as well as a unique transitional forest that contains a mixture of both mesic and dry conifer species [70]. These findings, however, contrast with our findings on mandibular morphospace, as both species display similar mandible shapes. We propose that plasticity may have a more critical role in these morphological traits and allows for greater convergence in muscle attachment and mechanical levers in a shared environment between these two squirrel species. Our findings add to previous studies in the *Tamiasciurus* hybrid zone that niche partitioning between red squirrels and Douglas squirrels is non-existent [45, 51].

Our findings also do not support expectations of the ecological competitive exclusion theory that species with shared resource requirements cannot stably coexist in an environment [71, 72]. Character displacement is one mechanism that allows sympatric species to reduce competition and coexist [8, 73–75], as shown in the feeding apparatus of other species in secondary contact [27]. However, our study does not show evidence for character displacement because the ecomorphological divergence in sympatry between the two squirrel species is not greater than the differences between them in allopatry. This finding is interesting in that these species are known to compete for territory access in sympatry [42, 51]. Interestingly, rather than convergence or character displacement, we find reduced bite force strength in both Douglas squirrels and red squirrels within the hybrid zone compared to their respective allopatric populations. However, the differences in bite force strength between Douglas squirrels and red squirrels remained relatively similar, suggesting that niche partitioning may occur in the hybrid zone, contrary to existing hypotheses for this system [45]. Despite the persistence of differences between Douglas squirrels and red squirrels within the hybrid zone, we also

observe that both species had lower bite force in the sympatric hybrid zone than in allopatry. For red squirrels, this shift may be associated with plasticity, as we would expect that they experience a softer diet in the hybrid zone than the allopatric range; however, we would expect the dietary challenge in the hybrid zone to be greater for Douglas squirrel compared to their allopatric range, yet they demonstrated a decline in bite force. The cause of this shared decline in bite force within the hybrid zone merits exploration in future studies. Another consideration for the lack of character displacement between sympatric red squirrels and Douglas squirrels is the presence of viable hybrids. Hybridization can have a dampening effect on character displacement between co-occurring and competing species where hybrids are viable and possess intermediate phenotypes [52]. However, this is not consistent with the hybrid phenotypes observed here, as hybrids phenotypically mirrored red squirrels rather than being intermediate between the parental species.

## Impact of hybridization on ecomorphologies

We find that ecomorphologies of hybrid squirrels follow different general expectations for hybrid phenotypes, including phenotypes that are more like one of the parental species, as well as phenotypes that span across the range of phenotypic variation of both parental species. BFQ, sagittal suture, and premaxillofrontal sutures in hybrids are more like sympatric red squirrels than Douglas squirrels. One explanation for why hybrid morphologies resemble red squirrel morphologies more closely and have greater morphological variation than Douglas squirrel is the genetic dominance of red squirrel alleles underlying these morphological traits. This is consistent with the lack of correlation between genetic ancestry proportions in hybrids and their incisor strength index or their suture complexity, suggesting further evidence that non-additive genetic variation, such as dominance or environmental effects, contributes to hybrid morphologies. However, a confounding factor that can also produce this pattern is the ecological selection for hybrid morphologies that more closely resemble red squirrel morphologies. Additional studies are needed to distinguish between the effects of genetic effects and selective advantages of red squirrel phenotypes. Addressing this possibility of competition may also shed light on the seemingly stable coexistence of the co-occurring species within the hybrid zone.

In contrast to the incisor strength index and the sagittal and premaxillofrontal sutures, the mandible shapes of hybrids and parental species located within the hybrid zone greatly overlap. One possibility for the similarity in mandible shape between hybrids and both parental species is that plasticity may have a more substantial effect on bony structures of the mandible than tooth shape [76]. Mandibles can be remodeled by their constant interactions with muscles and surrounding tissues, particularly when food hardnesses are variable between populations [23, 77]. In the *Tamiasciurus* hybrid zone, hybrids and parental species are found in syntopy and gnaw on cones from the same conifer species [42, 51]. Thus, it is conceivable that the shared biomechanical demands on the mandibles of both squirrel species and their hybrids in the hybrid zone, combined with the plasticity of the mandible have led to the overlap in mandibular shape.

Biologists have spent considerable effort in studying both the evolution of craniomandibular ecomorphogical traits, as well as hybrid zones across different mammalian taxa [78–82]. However, very few studies have combined these areas of research in trying to understand the consequences of admixture on trophic morphologies in mammals. One exception is the well-studied hybrid zone in central Europe between subspecies of house mice (*Mus musculus domesticus and M.m.musculus)* [79]. The phenotypic consequences of hybridization include heterosis, enhanced hybrid phenotypes, of the mandible and skull in hybrids [23, 83]. Despite

the impressive breadth of research on craniomandibular traits in house mice hybrids, including developmental stability, genetic architecture, and gross morphology [24, 84–86], existing studies are mostly laboratory-based and not necessarily true to ecology. In addition, the similar generalist diets of both mice subspecies can make it challenging to interpret the adaptive significance of morphological divergence. Our study system on *Tamiasciurus* squirrels presents contrasting virtues, and drawbacks to the house mice system as the adaptive significance of morphological variation are more apparently linked to the divergent ecologies between the two squirrel species. However, the larger body size of the squirrels and lower fecundity limit the types of experimental studies that can be conducted to thoroughly understand the developmental and genetic aspects of craniomandibular morphology.

Divergent ecological selection, interspecific competition, and hybridization are all ecological processes that can drive resource partitioning between closely related species and, ultimately, divergence in ecomorphologies. Our findings suggest that divergent selection associated with dietary differences in cone hardness has been an important driver of ecomorphological divergence between *T. douglasii* (soft-cone consumer) and *T. hudsonicus* (hard-cone consumer). Further experimental work, including dietary manipulation, is needed to examine the importance of plasticity. Through our study, we show that significant differences in ecomorphological traits are still observable in closely related taxa within a taxonomic group, like sciurids, that show a high degree of morphological conservatism. Finally, our use of fine-scale morphological data, such as bite force and suture complexity, provides additional evidence that high-resolution continuous traits are preferable over coarse traits for correctly detecting divergence instead of convergence [87].

## Materials and methods

### Specimen selection

To assess morphological differences between red squirrels, Douglas squirrels, and hybrids, we used a sample of 70 museum specimens across five groups (S1 Table): allopatric red squirrels (n = 14), hybrid zone red squirrels (n = 13), hybrids (n = 19), hybrid zone Douglas squirrels (n = 10), and allopatric Douglas squirrels (n = 14). Some specimens had varying degrees of damage and were excluded from some analyses. Final sample numbers for analyses are shown in the respective figures for those analyses. All specimens in this study are housed at the Burke Museum of Natural History (S1 Table; see S2 Fig for representative examples). Allopatric specimens were selected such that they were outside the hybrid zone (see S1 Table; Fig 1). The selection of specimens from within the hybrid zone was based on previous studies of the *Tamiasciurus* hybrid zone [42, 43]. We also used previously published estimates of admixture proportions for these same specimens [43, 51] based on the program STRUCTURE, an unsupervised clustering method implemented in a Bayesian framework. The estimated individual admixture proportions (Q) represent the fraction of an individual's genome coming from a given population [88] and were used to assign individuals to different genotypic classes (following assignment classes of [89]). We assigned individuals as either pure red squirrel (Q ≥ 0.90), pure Douglas squirrel (Q ≤ 0.10), or hybrid (0.10 < Q < 0.90).

### Bite force estimation

Maximal bite force was estimated from complete incisors using an incisor strength index ($Z_i$) of $Z_i = ((\text{antero-posterior length of incisor})^2 \times (\text{medio-lateral width of incisor}))/6$ (as detailed in [27]. This measure is strongly correlated with maximum *in vivo* bite force across Rodentia ($(\log_{10}(\text{bite force}) = 0.566 \log_{10}(Z_i) + 1.432$, $R^2 = 0.956$, [27]), and can be used to estimate maximal bite force for museum specimens where *in vivo* bite force data are unavailable. The

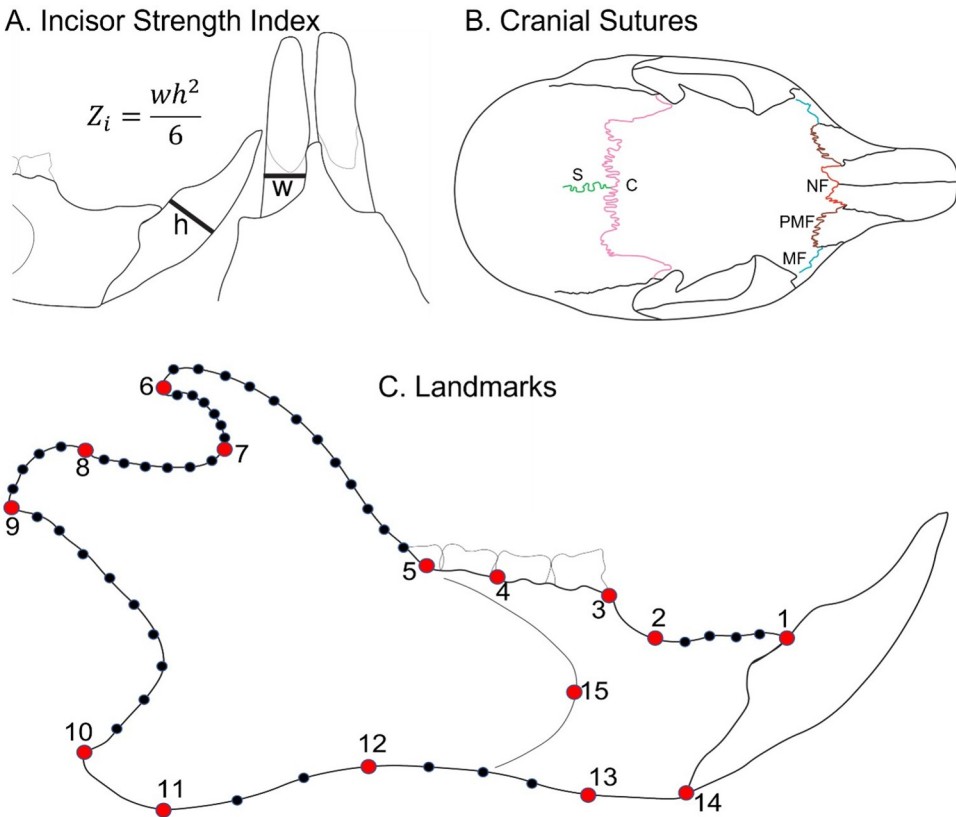

**Fig 7.** (A) Measurements of incisors in lateral and occlusal views and equation of the incisor strength index (labeled Zi) used as a proxy for bite force. (B) Schematic of cranial sutures used in this study with the nasofrontal suture (labeled NF) in red, the premaxillofrontal suture (labeled PMF) in brown, the maxillofrontal suture (labeled MF) in yellow, the coronal suture (labeled C) in pink, and the sagittal suture (S) in green. (C) Position of landmarks (large red circles) and semilandmarks (small black circles) on the mandible in lateral view. Descriptions of landmarks and semilandmarks are provided in S2 and S3 Tables.

anteroposterior length and the medio-lateral width of the lower right incisor of each specimen was measured using a Dino-Lite Edge AM7915 digital microscope by a single investigator (DMP) (Fig 7A). Because bite force scales allometrically with body size [29, 30, 90], bite force was converted to a bite force quotient (BFQ) using the residuals of the regression between $\log_{10}$(bite force) and $\log_{10}$(body mass) for all squirrel specimens in this study (following the method of [29]), with the regression (($\log_{10}$(bite force) = 0.3849 $\log_{10}$ (body mass) + 0.6970, $R^2$ = 0.419)) for this dataset. Bite force data were analyzed with an ANCOVA (using sex as a covariate) in RStudio version 1.4.1103 [91]. When ANCOVA results were significant, post-hoc pairwise comparisons were performed to test for differences among squirrel groups. To test for the potential effects of hybrid admixture on bite force estimation, we used a linear least squares regression of bite force and admixture score. We predicted that (1) red squirrels have a stronger BFQ than Douglas squirrels when compared allopatrically, (2) have more similar BFQ when compared sympatrically, and (3) hybrids have a phenotype intermediate between the sympatric parental species.

## Suture morphology

We quantified a complementary measure of bite force and skull suture complexity for all museum specimens with uninterrupted sutures of intact skulls. We measured five suture lines

(coronal, sagittal, nasofrontal, maxillofrontal, and premaxillofrontal sutures; Fig 7B) for each specimen. To do this, we photographed the dorsal view of the cranium from a fixed distance using a Canon EOS Rebel T6i. We then made vector graphics of sutures in Inkscape v0.93 (inkscape.org) by tracing the total length of the path of each suture on the images. We chose the right side of the crania to measure the maxillofrontal and premaxillofrontal sutures unless these were damaged, in which case, we measured the sutures on the left side. The complexity of each suture was quantified as a length ratio (LR), whereby the total length of the suture path is divided by the linear distance between the start and end [35, 92, 93]. Suture complexity data were analyzed with an ANCOVA, with sex as a covariate. When ANCOVA results were significant, post-hoc pairwise comparisons were performed to test for differences among squirrel groups. To test for a possible multivariate effect of suture complexity, all five sutures were also analyzed together using a principal component analysis (PCA). We used ANOVAs to test differences in PC1 and PC2 scores among squirrel groups. Finally, to test for the potential effects of hybrid admixture on suture complexity, we performed linear least squares regressions of suture complexity and admixture score for all five sutures. We predicted that (1) red squirrels have more complex sutures than Douglas squirrels when compared allopatrically, (2) have similarly complex sutures when compared sympatrically, and (3) hybrids have a phenotype intermediate between the sympatric parental species.

## Geometric morphometrics

To test for morphological divergence and morphospace overlap among the squirrel groups studied, we performed a geometric morphometric analysis of lower jaw shape including all complete specimens. The mandibles were photographed in lateral view on the same plane at a fixed distance using a Canon EOS Rebel T6i camera. For each specimen, 15 landmarks and 45 semilandmarks were digitized in TpsDig2 software [94]. The landmarks scheme was adapted from schemes used for squirrels [39] and hybridizing mice [24]. The semilandmarks used are equidistant points resampled to the lowest number, properly encapsulating the shape of the curve from homologous curves (Fig 7C, Landmark Description in S2 and S3 Tables). Landmark data were imported and analyzed using the R package geomorph v. 4.0.1 [95]. Semilandmarks were registered as sliders and slid according to the bending energy criterion [96]. Landmarks and semilandmarks were subjected to Generalized Procrustes Analysis (GPA) [97, 98]. A PCA was performed on the GPA data to explore group differences in shape data. A Monte Carlo randomization test run in biostats [99] was used to assess axes for which the amount of variation represented is significantly different from the null hypothesis. Procrustes ANOVAs were performed on these data in geomorph to assess the relationships between shape, size, sex, and group, with size represented by the log10-transformed centroid sizes from GPA. Pairwise comparisons were generated from these data in geomorph. We predicted that allopatric red squirrels would occupy a distinct morphospace from allopatric Douglas squirrels, that sympatric red squirrels and Douglas squirrels would have similar morphospace, and hybrids occupy an intermediate, and potentially overlapping, morphospace between the parental species.

Morphospace overlap between groups was analyzed using nicheROVER [100] using PC1 and PC2 scores, which represent the shape, and log10-transformed centroid size, which represents size. This analysis provides the probabilities of overlap between squirrel group pairs by taking advantage of a method developed to analyze multidimensional niche space overlap [101, 102]. Distributions were generated from the posterior distributions of 3,000 permutations of overlap probability. We predicted that for each species, the allopatric squirrels would have a high probability of overlap with hybrid zone conspecifics but low overlap with

heterospecific populations in allopatry, whereas hybrids would have a moderate to high probability of overlap with the other four squirrel groups. To test for the potential effects of hybrid admixture on mandible shape, we used a linear least squares regression of bite force and admixture score.

## Supporting information

**S1 Fig. Analysis of overlap probability of mandible geometric morphometric principal components 1 and 2 representing shape and log10-transformed centroid size representing size.** Data shown are a posterior histogram of 3000 iterations of overlap metric calculations, whereby the x-axis of each pairwise comparison is the range of overlap probabilities in percentage and the y-axis is frequency of that probability having been observed in calculation. Comparisons are for morphospace overlap between the squirrel group in the grid row on the squirrel group in the grid column. Mean overlap probability is shown as a solid black line and 95% confidence intervals are shown as dashed lines for all pairs.
(DOCX)

**S2 Fig. Representative examples for hybrid zone specimens.** (A) Douglas squirrel cranium from specimen UWBM 20809. (B) Douglas squirrel left mandible from specimen UWBM 20809, image is flipped. (C) Hybrid squirrel cranium from specimen UWBM 21078. (D) Hybrid squirrel right mandible from specimen UWBM 21078. (E) Red squirrel cranium from specimen UWBM 21034. (F) Red squirrel left mandible from specimen UWBM 21034, image is flipped.
(DOCX)

**S1 Table. Specimen museum IDs and locations.**
(XLSX)

**S2 Table. Homologous landmark descriptions.**
(DOCX)

**S3 Table. Semilandmark assignment and descriptions.**
(DOCX)

## Acknowledgments

We thank Jeff Bradley (University of Washington) for help and access to specimens in the University of Washington Burke Museum collections. We additionally thank Alec Sheets (The Ohio State University) for feedback and discussion on earlier versions of this manuscript.

## Author Contributions

**Conceptualization:** Dylan M. Poorboy, Andreas S. Chavez.

**Data curation:** Dylan M. Poorboy.

**Formal analysis:** Dylan M. Poorboy.

**Investigation:** Dylan M. Poorboy.

**Methodology:** Dylan M. Poorboy, Jonathan J.-M. Calède, Andreas S. Chavez.

**Project administration:** Jonathan J.-M. Calède, Andreas S. Chavez.

**Resources:** Jonathan J.-M. Calède, Andreas S. Chavez.

**Supervision:** Jonathan J.-M. Calède, Andreas S. Chavez.

**Validation:** Jonathan J.-M. Calède, Andreas S. Chavez.

**Visualization:** Dylan M. Poorboy, Jonathan J.-M. Calède.

**Writing – original draft:** Dylan M. Poorboy, Jonathan J.-M. Calède, Andreas S. Chavez.

**Writing – review & editing:** Dylan M. Poorboy, Jonathan J.-M. Calède, Andreas S. Chavez.

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
