## [Decision Letter · Decision Letter 0]

8 Jan 2023

PONE-D-22-29005Craniodental divergence associated with bite force between hybridizing pine squirrels (Tamiasciurus)PLOS ONE

Dear Dr. Poorboy,

Thank you for submitting your manuscript to PLOS ONE. After careful consideration, we feel that it has merit but does not fully meet PLOS ONE’s publication criteria as it currently stands. Therefore, we invite you to submit a revised version of the manuscript that addresses the points raised during the review process.

 Please present more details about the hybrid zone. Both reviewers asked for information about available ressources and composition of this forest as it may also have an impact on the observations. Please also consider the various comments of the reviewers, asking for some precisions about your results. I also remind you that the journal policy requires authors to make all data underlying the findings described in their manuscript fully available without restriction.

We look forward to receiving your revised manuscript.

Kind regards,

Cyril Charles

Academic Editor

PLOS ONE

Journal Requirements:

4. We note that Figure 1 in your submission contain map images which may be copyrighted. All PLOS content is published under the Creative Commons Attribution License (CC BY 4.0), which means that the manuscript, images, and Supporting Information files will be freely available online, and any third party is permitted to access, download, copy, distribute, and use these materials in any way, even commercially, with proper attribution. For these reasons, we cannot publish previously copyrighted maps or satellite images created using proprietary data, such as Google software (Google Maps, Street View, and Earth). For more information, see our copyright guidelines: http://journals.plos.org/plosone/s/licenses-and-copyright.

Reviewers' comments:

Reviewer's Responses to Questions

**Comments to the Author**

1. Is the manuscript technically sound, and do the data support the conclusions?

Reviewer #1: Yes

Reviewer #2: Yes

2. Has the statistical analysis been performed appropriately and rigorously? 

Reviewer #1: Yes

Reviewer #2: Yes

3. Have the authors made all data underlying the findings in their manuscript fully available?

Reviewer #1: No

Reviewer #2: Yes

4. Is the manuscript presented in an intelligible fashion and written in standard English?

Reviewer #1: Yes

Reviewer #2: Yes

5. Review Comments to the Author

Reviewer #1: This is in my opinion a very straightforward, sound and complete study on a nice model system, combining several morpho-functional characters within a hybridization zone context, which is rare enough to make it particularly interesting. I think it is essentially publishable as is, and have only a few minor comments and suggestion, with one caveat however, which is that I could not have access to the underlying data or code. As far as I can see, the methods used, results obtained and their interpretation appear sound, but it should certainly be double checked that data is actually made available at acceptance.

With that caveat aside, here are my comments and suggestions, which I hope the authors might find useful and interesting.

Introduction:

- Line 70 the authors state that "traits associated with foraging are routinely under strong selection". Although this makes intuitive sense, reference [26] by itself seems a bit light to back this up, maybe add a few others (one can think obviously of the Grant and Grant finches studies, but there are also probably a few others out there on rodents/small mammals).

- Lines 74-77 and throughout the paper: The authors did a great job in explaining the incisor strength index, which in my opinion was not very clearly described in the original paper (ref [27]). However, I would raise one caveat with this index, which is that, as far as I know, it was not tested against in vivo bite forces in an intra-specific context. I realise the current study is not per se at the intra-specific level, but it is closer to that than to the broad taxonomic level used in [27]. I would therefore suggest that the authors either mention intra-specific studies which I may not be aware of, or warn the readers that this index may not closely match actual bite forces.

- Line 86: Have divergent selection and inter-specific competition been demonstrated before in those squirrels? If so, please cite the appropriate studies.

- Lines 94-100: Here, and in other places throughout the manuscrot, I came to wonder about the composition of the hybrid zone forest. This information only came towards the end, in the Discussion (lines 329-333). I think it would be better for readers to have a brief description of the community of trees in the HZ directly in the Introduction.

- On a somewhat related note, it would also be nice to have pictures of the cones, and possibly pictures of some typical skull for each species. Perhaps as insets in Fig. 1? Or as supplementary material?

- Lines 100-104, and later in the manuscript (lines 275-279...): As far as I can see in refs [40, 45] the link between sagittal crest dimensions and bite force was not actually tested. Here the authors could do it, as they have a supposedly good proxy for bite force (incisor strength index), and specimens for which sagittal crest was apparently measured in study [40] (according to Mat & Met specimens studied here were selected from specimens in [40]). I believe a correlation between these two traits would be a nice confirmation of the importance of the sagittal crest in biting function.

- Line 103: [40,45) should be [40,45] I guess.

- Line 135: I think the phrasing of hypothesis (3) is confusing. If the two species have "intense competition for shared food resources" it seems to imply that they use the same resource, in which case one would expect convergent phenotypes to be better able to use that shared resource. On the other hand character displacement would imply that each species specializes on a different resource (or, for instance, get it from different places), leading to different phenotypes. Perhaps the sentence was meant as "character displacement due to past competition", rather than ongoing competition?

- Line 137: I do not see why hypothesis (3) would be expected. Is there any previous study showing that? Ref 40 seems to show intermediate phenotypes rather than morphologies "outside the range of parental species". Furthermore, intermediate phenotypes would, it seems to me be the more obvious "null hypothesis". Finally, this hypothesis (3) is contrary to hypothesis (2) of the Suture Morphology section in the Mat & Met. Authors should be consistent in their expectations.

Results:

- Line 142: As Results came before Mat & Met, the "BFQ" was not introduced before. Authors should perhaps mention it in the Introduction, for instance in the paragraph in which they introduce the incisor strength index. Or at least in the results mention the Bite Force Quotient before using an abbreviation.

- Fig. 4 and its legend. It seems to me that the amount of deformation in the deformation grids might have been amplified (i.e. do not reflect actual variation in specimens). I understand that it is quite standard to do so, but if that is the case, please mention it in the caption of the figure, with the amount of amplification.

- Line 202: Not sure what "significant axes" in a PCA is supposed to mean. Biologically relevant? Decent signal to noise ratio? I think the authors might need to expand on that in the Mat & Met section lines 500-502.

- Line 204-206 (and 237), the authors refer to the morphology of the "dentary". As far as I am aware, the mammalian madible is constituted exclusively by the dentary bone. However, elsewhere in the MS, the authors refer generally to the morphology of the mandible. Does this have a special meaning here? If so, please specify what is meant. If not, please be consistent across the manuscript, by refering only to the mandible.

- Line 214-215: I think the authors did well to check the effect of size and sex, and I would be curious to see similar analyses run on i) the Zi and ii) on the PC1 and 2 for suture complexity.

Discussion:

- As mentionned before it would be interesting to test sagittal crest data against Zi/BFQ and even maybe suture complexity.

- Line 300-301: Do they really open "during" the fires? or right after?

- Line 338: "Not apparent or very subtle" seems almost synonymous. Maybe "non-existent" was meant?

- Lines 339-345: This is quite interesting, so were there any previous studies in that hybrid zone showing that one species had an actual impact on the other species' population density (i.e. direct evidence for competition)? From my own experience (sorry for self advertising) I know at least one other example of Murid rodents living in syntopy without evidence of competition. Our interpretation was that the abundance of resources is enough to sustain thriving populations of both species (see e.g. Bauduin et al 2013 https://doi.org/10.1139/cjz-2012-0286, Ginot et al 2018 https://doi.org/10.1139/cjz-2017-0243), could that also be the case in those hybrid zone forests?

- Lines 346-357: This is perhaps the most intriguing result here. I wonder what could be the role of that third pine species here. Would love to see future studies about that.

- Line 369: Here incisor strength index is mentioned, but what is shown in Fig. 2 for example is BFQ, not Zi.

- Lines 398-400: It seems the sentence is missing a part here. "Despite [such and such]..." then what?

Materials and Methods:

- data not available so far

- Line 442: if I am not mistaken, this regression is across species means. Individual values in [27] were not used to test at the intraspecific scale the relation between Zi and in vivo bite force.

- Line 446-448: I think it would be nice to mention somewhere the results of this regression.

- Line 449: there is a missing parenthesis.

- Line 453: Red squirrels bite harder, but are they of the same size? Is that statement true for size corrected bite forces?

Reviewer #2: In this very interesting study, the authors wanted to evaluate the ecomorphological variations of some skull traits in allopatric and sympatric populations of squirrels involving hybridizations. They nicely compared and investigated different morphological traits associated with bite forces, in relation to food resources having different hardnesses.

I have only two main concerns:

1 – The kind of food/plant found in the hybrid zones and its associated mechanical properties should be more clearly presented in the introduction and in the discussion to better understand the morphological patterns observed in the sympatric populations.

2 - The interpretation of the results concerning populations from the hybrid zone can appear as speculative at some points in the discussion. The putative roles of genetic processes, phenotypic plasticity and selection should be tempered, especially for contradictory results (e.g. HZ Douglas squirrels).

Most of my other comments and suggestions are listed in the pdf file.

6. PLOS authors have the option to publish the peer review history of their article (what does this mean?). If published, this will include your full peer review and any attached files.

Reviewer #1: **Yes: **Samuel Ginot

Reviewer #2: **Yes: **Helder GOMES RODRIGUES

---

## [Author Response · Author response to Decision Letter 0]

19 Mar 2023

Editor Comments:

Please present more details about the hybrid zone. Both reviewers asked for information about available resources and composition of this forest as it may also have an impact on the observations. Please also consider the various comments of the reviewers, asking for some precisions about your results. I also remind you that the journal policy requires authors to make all data underlying the findings described in their manuscript fully available without restriction.

We have modified the manuscript in response to the comments made by the reviewers. For an itemized list of edits made and responses to reviewers please see below. We can and will provide accessibility to all data associated with this study should our manuscript be accepted for publication.

Journal Requirements: 

We have checked that the submitted manuscript meets PLOS ONE’s style requirements.

We are prepared to provide repository information for all data used in this study should the manuscript be accepted for publication. If there is a preferred repository for data accessibility, please let us know and we would be happy to make our data available there. 

This research was entirely conducted using museum specimens acquired from the Burke Museum of Natural History and Culture and does not fall under any approval process or ethics committee review. 

4. We note that Figure 1 in your submission contain map images which may be copyrighted. All PLOS content is published under the Creative Commons Attribution License (CC BY 4.0), which means that the manuscript, images, and Supporting Information files will be freely available online, and any third party is permitted to access, download, copy, distribute, and use these materials in any way, even commercially, with proper attribution. For these reasons, we cannot publish previously copyrighted maps or satellite images created using proprietary data, such as Google software (Google Maps, Street View, and Earth). For more information, see our copyright guidelines: http://journals.plos.org/plosone/s/licenses-and-copyright.

We have edited the Fig 1 caption to include references for both the basemap data and the shapefile range maps used in displayed map. The caption now includes the statement: “Douglas squirrels (T. douglasii) and red squirrels (T. hudsonicus) range data reprinted from IUCN shapefiles [49,50]. Basemap reprinted from Terrain With Labels under a CC BY license, with permission from ESRI, original copyright 2016.”. 

Captions for Table S1, Table S2, Table S3, Fig S1, and the new Fig S2 have been added to the end of the manuscript.

We have modified the reference list to include new references added in response to editor and reviewer comments and reordered the list accordingly. In-text citations throughout the manuscript have been renumbered to reflect changes.

Reviewer Comments:

Reviewer #1: This is in my opinion a very straightforward, sound and complete study on a nice model system, combining several morpho-functional characters within a hybridization zone context, which is rare enough to make it particularly interesting. I think it is essentially publishable as is, and have only a few minor comments and suggestion, with one caveat however, which is that I could not have access to the underlying data or code. As far as I can see, the methods used, results obtained and their interpretation appear sound, but it should certainly be double checked that data is actually made available at acceptance.

With that caveat aside, here are my comments and suggestions, which I hope the authors might find useful and interesting.

Introduction:

-Line 70 the authors state that "traits associated with foraging are routinely under strong selection". Although this makes intuitive sense, reference [26] by itself seems a bit light to back this up, maybe add a few others (one can think obviously of the Grant and Grant finches studies, but there are also probably a few others out there on rodents/small mammals).

We have added citations for Grant and Grant 2006, Zelditch et al. 2017, and Wroe et al. 2005 to this statement to help reinforce the presented idea. Citations numbers for these papers are [27-29] in the revised manuscript.

- Lines 74-77 and throughout the paper: The authors did a great job in explaining the incisor strength index, which in my opinion was not very clearly described in the original paper (ref [27]). However, I would raise one caveat with this index, which is that, as far as I know, it was not tested against in vivo bite forces in an intra-specific context. I realise the current study is not per se at the intra-specific level, but it is closer to that than to the broad taxonomic level used in [27]. I would therefore suggest that the authors either mention intra-specific studies which I may not be aware of, or warn the readers that this index may not closely match actual bite forces.

We have added a new statement to address this potential problem, indicating that we do expect this metric to be a reliable proxy for comparing approximate biting strength but may not be applicable in estimating actual units of force. The text added to the introduction reads: “This index was developed from interspecific studies in deeper evolutionary time than is addressed herein; however, we expect it to be functional as a proxy measure of comparing relative biting strength, although it may not be directly translatable to units of force.”

- Line 86: Have divergent selection and inter-specific competition been demonstrated before in those squirrels? If so, please cite the appropriate studies.

We have integrated additional material on previous knowledge for this system into the introduction of the manuscript. The passage reads as: “The territorial behavior and overlapping niche of both parental species and their hybrids within the hybrid zone indicate that interspecific competition among all squirrel types is potentially strong, which may have important consequences on access to food resources and, ultimately, the evolution of craniodental morphologies associated with foraging.”

- Lines 94-100: Here, and in other places throughout the manuscript, I came to wonder about the composition of the hybrid zone forest. This information only came towards the end, in the Discussion (lines 329-333). I think it would be better for readers to have a brief description of the community of trees in the HZ directly in the Introduction.

We have added additional detail on the main conifers present in the HZ, primarily from the Smith 1968 and Fotis 2020 studies (citation numbers now 42 and 51 in the revised manuscript, respectively). The added statement reads as: “Within the hybrid zone, observational studies have shown that both parental squirrel species and their hybrids are syntopic and select similar food items in subalpine forests dominated by Engelmann spruce and subalpine fir [42,51], which are expected to be qualitatively soft cones. Hybrids can also backcross successfully with both parental species [43].”

- On a somewhat related note, it would also be nice to have pictures of the cones, and possibly pictures of some typical skull for each species. Perhaps as inserts in Fig. 1? Or as supplementary material?

We do not have our own pictures of the cones to make available as a figure. Such pictures also wouldn’t necessarily yield additional information regarding characteristics of hardness, which is expected to be more tactile than necessarily visual. We have made pictures of the crania and mandibles for representative individuals for the species and hybrids in the form of new Supplementary Fig S2.

- Lines 100-104, and later in the manuscript (lines 275-279...): As far as I can see in refs [40, 45] the link between sagittal crest dimensions and bite force was not actually tested. Here the authors could do it, as they have a supposedly good proxy for bite force (incisor strength index), and specimens for which sagittal crest was apparently measured in study [40] (according to Mat & Met specimens studied here were selected from specimens in [40]). I believe a correlation between these two traits would be a nice confirmation of the importance of the sagittal crest in biting function.

We agree that this analysis would be interesting, but it is beyond the scope of our available data. The nature of the sagittal crest in these taxa is primarily characterized as the presence of a prominent sagittal crest or a flattened but widened sagittal plateau. Means of measuring the crest would be better captured in a study that could use 3-dimensional data (which we unfortunately lack for these specimens) and dissection of the temporal muscle, as quantitative muscle attachment differences would be necessary in understanding how the observed sagittal crest and sagittal plateau differ. We have added an additional statement to the discussion stating “The relationship between greater biting forces, the sagittal suture, and sagittal crest size could be addressed in future studies accompanied by temporal muscle dissections to quantify muscle attachment space.” to emphasize the usefulness of continuing to explore this area. 

- Line 103: [40,45) should be [40,45] I guess.

We implemented the change suggested by the reviewer.

- Line 135: I think the phrasing of hypothesis (3) is confusing. If the two species have "intense competition for shared food resources" it seems to imply that they use the same resource, in which case one would expect convergent phenotypes to be better able to use that shared resource. On the other hand character displacement would imply that each species specializes on a different resource (or, for instance, get it from different places), leading to different phenotypes. Perhaps the sentence was meant as "character displacement due to past competition", rather than ongoing competition?

We have changed the statement of hypothesis 2 to read as “ecomorphologies between Douglas squirrels and red squirrels in sympatry within the hybrid zone are convergent, consistent with a shared ecology”. We have also added the sentence “Alternatively, convergent evolution may occur for these species due to the selective pressures of a shared food resource.” earlier in the introduction to setup the logic of this expectation given existing knowledge of these taxa.

The phrasing of predictions in material and methods have also been adjusted to mirror this expectation of convergence and now read as follows:

BFQ: “We predicted that (1) red squirrels have a stronger BFQ than Douglas squirrels when compared allopatrically, (2) have more similar BFQ when compared sympatrically, and (3) hybrids have a phenotype intermediate between the sympatric parental species.”

Suture Complexity: “We predicted that (1) red squirrels have more complex sutures than Douglas squirrels when compared allopatrically, (2) have similarly complex sutures when compared sympatrically, and (3) hybrids have a phenotype intermediate between the sympatric parental species.”

Geometric Morphometrics: “We predicted that allopatric red squirrels would occupy a distinct morphospace from allopatric Douglas squirrels, that sympatric red squirrels and Douglas squirrels would have similar morphospace, and with hybrids occupying occupy an intermediate, and potentially overlapping, morphospace between the parental species.”

Morphospace Overlap: “We predicted that for each species, the allopatric squirrels would have a high probability of overlap with hybrid zone conspecifics but low overlap with heterospecific populations in allopatry, whereas hybrids would have a moderate to high probability of overlap with the other four squirrel groups.”

- Line 137: I do not see why hypothesis (3) would be expected. Is there any previous study showing that? Ref 40 seems to show intermediate phenotypes rather than morphologies "outside the range of parental species". Furthermore, intermediate phenotypes would, it seems to me be the more obvious "null hypothesis". Finally, this hypothesis (3) is contrary to hypothesis (2) of the Suture Morphology section in the Mat & Met. Authors should be consistent in their expectations.

We have changed this hypothesis to read as follows: “hybrid morphologies are overlapping with or intermediate to either parental species due to genetic admixture between the parental species and a shared ecology.” to be more logically consistent with the current understanding of these taxa and the predictions made in materials and methods.

Results:

- Line 142: As Results came before Mat & Met, the "BFQ" was not introduced before. Authors should perhaps mention it in the Introduction, for instance in the paragraph in which they introduce the incisor strength index. Or at least in the results mention the Bite Force Quotient before using an abbreviation.

We have added the full wording for “bite force quotient” to the Results section to properly introduce it here. The results section now begin as: “The bite force quotient (BFQ) of the lower incisor differs significantly…”. 

- Fig. 4 and its legend. It seems to me that the amount of deformation in the deformation grids might have been amplified (i.e. do not reflect actual variation in specimens). I understand that it is quite standard to do so, but if that is the case, please mention it in the caption of the figure, with the amount of amplification.

We added additional detail to the figure legend to explain these deformation grids. They represent the extremes as graphically shown for each axis. The Fig 4 legend now reads as: “Fig 4. Principal component analysis of mandible shape in lateral view. Deformation grids for principal component axes accompany their respective axis and are graphical extremes from the principal component plot. Abbreviations are provided in Fig 2.”

- Line 202: Not sure what "significant axes" in a PCA is supposed to mean. Biologically relevant? Decent signal to noise ratio? I think the authors might need to expand on that in the Mat & Met section lines 500-502.

We have added clarifying detail of this method to the material and methods section and the results section as follows:

Results: “Monte Carlo randomization supported PC1 (p < 0.001) and PC2 (p < 0.001) as significant axes for the morphospace PCA, with all other axes being nonsignificant (p > 0.99), thus, only these axes were retained for subsequent analyses.”

Methods: “Monte Carlo randomization test run in biostats [99] was used to assess axes for which the amount of variation represented is significantly different from the null hypothesis.”

- Line 204-206 (and 237), the authors refer to the morphology of the "dentary". As far as I am aware, the mammalian mandible is constituted exclusively by the dentary bone. However, elsewhere in the MS, the authors refer generally to the morphology of the mandible. Does this have a special meaning here? If so, please specify what is meant. If not, please be consistent across the manuscript, by referring only to the mandible.

We have changed the terminology here to “mandible” to be consistent with the rest of the manuscript. We also changed a use of “dentaries” in the Geometric Morphometrics method section to “mandibles” to maintain this consistent terminology.

- Line 214-215: I think the authors did well to check the effect of size and sex, and I would be curious to see similar analyses run on i) the Zi and ii) on the PC1 and 2 for suture complexity.

We have now added this for PC1 and PC2 for suture complexity and adjusted the original reported ANOVA. The modified results read as “An ANCOVA reveals a significant difference in PC1 scores among squirrel groups (Group: p < 0.0001, F4,45 = 13.38; sex: p =0.002, F1,45 = 10.47; size: p = 0.42, F1,45 = 0.67), but no significant difference is present for PC2 among groups (Group p = 0.528, F4,45 = 0.81; sex: p =0.97, F1,45 = 0.002; size: p = 0.38, F1,45 = 0.80).”. We have not added the analyses for Zi, as our analysis of BFQ is already corrected for size and uses sex as a covariate. 

Discussion:

- As mentioned before it would be interesting to test sagittal crest data against Zi/BFQ and even maybe suture complexity.

Please refer to comment addressing original lines 100-104.

- Line 300-301: Do they really open "during" the fires? or right after?

We changed the wording to “following” to more accurately describe the timing of serotinous cone opening in the event of a fire.

- Line 338: "Not apparent or very subtle" seems almost synonymous. Maybe "non-existent" was meant?

We agree with non-existent being the best phrasing here and have made the change suggested by the reviewer. The passage now reads as: “Our findings add to previous studies in the Tamiasciurus hybrid zone that niche partitioning between red squirrels and Douglas squirrels is either not apparent or very subtlenon-existent”.

- Lines 339-345: This is quite interesting, so were there any previous studies in that hybrid zone showing that one species had an actual impact on the other species' population density (i.e. direct evidence for competition)? From my own experience (sorry for self advertising) I know at least one other example of Murid rodents living in syntopy without evidence of competition. Our interpretation was that the abundance of resources is enough to sustain thriving populations of both species (see e.g. Bauduin et al 2013 https://doi.org/10.1139/cjz-2012-0286, Ginot et al 2018 https://doi.org/10.1139/cjz-2017-0243), could that also be the case in those hybrid zone forests?

We have added an additional statement that reads as: “This finding is interesting in that these species are known to compete for territory access in sympatry [42,51].” Behaviorally, these squirrels are fierce defenders of these territories, so resources are not expected to be abundant enough to mitigate selection.

- Lines 346-357: This is perhaps the most intriguing result here. I wonder what could be the role of that third pine species here. Would love to see future studies about that.

Indeed, we propose that future studies should be conducted to assess the possibility of genetic dominance or a competitive advantage of the red squirrel phenotype in hybrids (as all specimens in this study were adult size and survived to maturation) causing the skew. See the passage “Additional studies are needed to distinguish between the effects of genetic effects and selective advantages of red squirrel phenotypes. Addressing this possibility of competition may also shed light on the seemingly stable coexistence of the co-occurring species within the hybrid zone.”

- Line 369: Here incisor strength index is mentioned, but what is shown in Fig. 2 for example is BFQ, not Zi.

We have made the textual change here to “BFQ” to better represent what was specifically tested for in our analyses, as suggested. 

- Lines 398-400: It seems the sentence is missing a part here. "Despite [such and such]..." then what?

We have fixed this sentence to include the entirety of the intended point. The sentence now reads as: “Despite the impressive breadth of research on craniomandibular traits in house mice hybrids, including developmental stability, genetic architecture, and gross morphology [24,84-86], existing studies are mostly laboratory-based and not necessarily true to ecology. In addition, the similar generalist diets of both mice subspecies can make it challenging to interpret the adaptive significance of morphological divergence.” 

Materials and Methods:

- data not available so far

Data for this study can and will be made available at the time of publication in accordance with the PLOS data policy. 

- Line 442: if I am not mistaken, this regression is across species means. Individual values in [27] were not used to test at the intraspecific scale the relation between Zi and in vivo bite force.

We have added a short caveat statement regarding the potential shortcomings of this method to the introduction, in addressing an earlier comment. We believe it is still a useful metric for comparing biting strength but agree that it may not be directly translatable to real force units. The added statement to the introduction reads as “This index was developed from interspecific studies in deeper evolutionary time than is addressed herein; however, we expect it to be functional as a proxy measure of comparing approximate biting strength, albeit may not be directly translatable to units of force.”

- Line 446-448: I think it would be nice to mention somewhere the results of this regression.

We have added the regression equation used for calculating BFQ to these methods as log10(bite force) = 0.3849 log10 (body mass) + 0.6970, R2=0.419. This equation is dataset specific. 

- Line 449: there is a missing parenthesis.

We implemented the change suggested by the reviewer.

- Line 453: Red squirrels bite harder, but are they of the same size? Is that statement true for size corrected bite forces?

We adjusted this sentence to indicate that our prediction is really for greater BFQ, which accounts for size. Red squirrels are typically slightly larger than Douglas squirrels, but we predicted that a difference in strength would be maintained even with the size correction made by converting the data to BFQ. 

Reviewer #2: In this very interesting study, the authors wanted to evaluate the ecomorphological variations of some skull traits in allopatric and sympatric populations of squirrels involving hybridizations. They nicely compared and investigated different morphological traits associated with bite forces, in relation to food resources having different hardnesses.

I have only two main concerns:

1 – The kind of food/plant found in the hybrid zones and its associated mechanical properties should be more clearly presented in the introduction and in the discussion to better understand the morphological patterns observed in the sympatric populations.

2 - The interpretation of the results concerning populations from the hybrid zone can appear as speculative at some points in the discussion. The putative roles of genetic processes, phenotypic plasticity and selection should be tempered, especially for contradictory results (e.g. HZ Douglas squirrels).

Most of my other comments and suggestions are listed in the pdf file.

Page: 8

Line 40: To be more specific, hybrids more importantly overlap with HZ red squirrels than with HZ Douglas squirrels (taking into account all the morphological traits).

We implemented the change suggested by the reviewer. The sentence now reads as: “Furthermore, we find that hybrids display morphologies that overlap with sympatric hybrid zone red squirrels but not with sympatric hybrid zone Douglas squirrels”

Line 40-42: You probably mean "not significantly different".

We implemented the change suggested by the reviewer. The sentence now reads as: “We find these sister squirrel species are different in bite force and suture complexity in both allopatry and sympatry and that mandible shape changes with the expected hardness of accessed food items, but not directlyis not significantly different between species.”

Line 50: different

We implemented the change suggested by the reviewer.

Line 51: Add precision or examples please.

We have added specification for morphological and behavior divergent selection, consistent with the studies cited to make this point. The sentence now reads as: “At shallow timescales, ecological different processes, including divergent ecological selection of morphology and behavior, hybridization, and interspecific competition, can be important initiators of ecomorphological diversity [1-3].”

Line 54: different

We implemented the change suggested by the reviewer.

Line 58-59: I don't understand why phenotypic convergence would not involve evolution of character in a context of similar selective pressures (that might be specific to the hybrid zone).

We rephrased this sentence to make it clearer that the contrast is between divergent evolution driven by character displacement and convergent evolution driven by shared ecological pressures. The sentence now reads as: “Furthermore, interspecific competition within hybrid zones can potentially drive the divergent evolution of through character displacement [7-9] or, in contrast, phenotypic convergence driven by shared ecological pressures [10-12].”

Line 70-72: There is a problem of transition with the last paragraph on hybrid zones. The present paragraph should be put after a brief presentation of ecological issues of Tamiasciurus in hybrid zones (or within this presentation).

We have made several adjustments of information presentation throughout the introduction that should make this more fluid.

Line 80: Mandible shape is also a good predictor of bite forces (at least in some rodents). See Ginot et al., 2019, Journal of Experimental Biology.

We have added the additional reference of Ginot et al. 2019 (citation number 38 and some brief explanation of the predictive power of mandible shape for bite force to this passage. The sentence now reads as: “Finally, mandible shape is another demonstrated predictor of bite force [38] and may influence masticatory muscle attachment and mechanical levers, with the expectation that more durophagous animals will have larger or more robust mandibular processes and higher mechanical lever advantages at the incisors [28, 39-41].”

Line 81: masticatory

We implemented the change suggested by the reviewer. The change is reflected in the quoted sentence addressing original line 80 above.

Line 82: Precise which processes (not all of them).

We changed the text to specify “mandibular” processes, as the work of Casanovas-vilar and van Dam (2013) found all mandibular processes to be well developed for tree squirrels. This change is reflected in the quoted sentence addressing original line 80 above.

Line 83: Add "at incisors".

We implemented the change suggested by the reviewer. This change is reflected in the quoted sentence addressing original line 80 above.

Line 84-85: Precise Tamiasciurus.

We implemented the change suggested by the reviewer for the first use case with Tamiasciurus. Douglasii. The second case is kept as T. hudsonicus.

Line 91-93: Add a reference please.

We added the Smith 1968 and 1981 (citation numbers 42 and 45, respectively) references here and modified the sentence to read as follows: “Tamiasciurus squirrels are strongly associated with coniferous forests, feed primarily on conifer seeds by mechanically removing cone scales with their mouth, are mostly asocial, and will vigorously defend territories to protect their centralized cache of conifer cones [42,45].”

Line 100-103: You should briefly mention that the presence of a sagittal crest is related to the development of the temporal muscle, which has a high contribution in the production of bite force.

We have added additional detail for the suggested connection between the presence/absence of sagittal crest and the development of the temporal muscle. The new sentence reads as: “This difference in sagittal crest development is expected to drive a difference in produced bite force for these species, as it relates to muscle attachment space for the temporalis muscle involved in the biting motion.”

Line 114-122: Does it exist any direct observations on the field regarding the behaviour and ecology of these squirrels from the hybrid zone?

We rephrased the information presented to indicate that the behaviors described are observed in the cited material within the hybrid zone. The new passage reads as: “Within the hybrid zone, observational studies have shown that both parental squirrel species and their hybrids are syntopic and select similar food items in subalpine forests dominated by Engelmann spruce and subalpine fir [45,51], which are expected to be qualitatively soft cones. Hybrids can also backcross successfully with both parental species [43]. The territorial behavior and overlapping niche of both parental species and their hybrids within the hybrid zone indicate that interspecific competition among all squirrel types is potentially strong, which may have important consequences on access to food resources and, ultimately, the evolution of craniodental morphologies associated with foraging.”

Line 124: Precise please: kind of food and hardness in the hybrid zone, and for each population if available?

We have added additional detail on the main conifers present in the HZ, primarily from the Smith 1968 and Fotis 2020 studies (citation number 45 and 51, respectively). These studies have described this HZ environment as being dominated by Engelmann spruce and subalpine fir, which are the prominent energetically valuable seed sources used by both species of squirrels. The new information is presented in the quote addressing the original line 114-122 above. 

 Line 128 – text deletion suggested.

We implemented the change suggested by the reviewer.

Line 169: "."

We implemented the change suggested by the reviewer.

Line 185: To be italicized.

We implemented the change suggested by the reviewer.

Line 192: It could have also been tested for mandible shapes (i.e. PC1 coordinates and centroid sizes).

We have added these analyses; however, neither was found to have a significant correlation. 

Newly added text for the results reads as “Admixture was also not significantly correlated with PC1 of the morphospace PCA (R2 = 0.08, p = 0.314) or centroid size (R2 = 0.004, p = 0.811).” The overall “Hybrid admixture regressions” passage of the results was also moved to come after the “Geometric Morphometrics” material to place the principal component and centroid size data in context.

Newly added to methods: “To test for the potential effects of hybrid admixture on mandible shape, we used a linear least squares regression of bite force and admixture score.”

Line 202-203: You should indicate the results of your MC randomization.

We have added the results for our MC randomization test, as requested. The added results read as: “Monte Carlo randomization supported PC1 (p < 0.001) and PC2 (p < 0.001) as significant axes for the morphospace PCA, with all other axes being nonsignificant (p > 0.99), thus, only these axes were retained for subsequent analyses.”. 

Line 206: text deletion suggested.

The suggested text deletion was for anatomical directions, which we would prefer to keep in the manuscript for a more accurate and detailed description. 

Line 207-209: Or "... a short ramus including a reduced coronoid process and a short angular process".

We implemented the change suggested by the reviewer. The modified passage reads as: “Positive PC2 values are associated with a short ramus, including a reduced coronoid process and a short angular process.”

Line 206-207: I would have said "an enlarged". You should also describe the higher coronoid process.

We implemented the change suggested by the reviewer. The modified passage reads as: “Specimens with more positive PC1 values have an anteroposteriorly shortened mandible, a shortened condylar process, an enlarged angular process, and a higher coronoid process.”

Line 211-212: Left part of virtual deformations are missing on PC2

We have resized this figure for resubmission to include the missing margins.

Line 215-216: For PC1 and PC2 only?

We made it more explicit that these results are for PC1 and PC2 alone. 

Line 229: You mean "no significant difference associated".

We tried to make our statement of no significant pairwise differences more clear here. The new statement reads as: “The Procrustes ANOVA indicates a significant effect of group on mandible shape (p = 0.001, F4,46 = 6.41); however, there were no significant pairwise differences associated with Procrustes data after correction for multiple testing.”

Line 258-262: This part should have been more relevant in the introduction.

We have polished writing throughout the introduction and rearranged information in response to other comments, which should make these points more relevant leading into the discussion.

Line 264-266: Ok, but this is one of your main conclusions. Here, you cannot state that red squirrels have different morphological traits indicating stronger bite force, but only BQF. You should present first your interpretations and then conclude.

We have changed the wording here from “bite force” to “BFQ” so as to not overinterpret the results.

Line 284: As well as biting associated stress/strains for premaxilla.

We have rephrased this sentence and added clarifying detail. It now reads as “These specific sutures are associated with tooth-bearing bones (premaxilla and maxilla) and are specifically expected to be more complex in response to increased masticatory strains [61].” 

Line 285-287: And a higher coronoid process (cf insertion of temporalis).

We implemented the change suggested by the reviewer. The new passage reads as “Allopatric red squirrels have a shorter mandibular corpus, a shorter condylar process, a longer angular process, and a higher coronoid process than allopatric Douglas squirrels.”

Line 286: text deletion suggested. 

We implemented the change suggested by the reviewer. 

Line 286: text deletion suggested. 

We implemented the change suggested by the reviewer.

Line 290: You probably mean angular and mainly coronoid processes.

We added specifics to make it clearer that well developed angular and coronoid processes are defining characteristics of nut-eating squirrels. The modified sentence reads as: “This pattern is consistent with the general pattern defining nut-eating squirrels in which they have been characterized as having a robust mandibular corpus, high ramus, and well-developed angular and coronoid processes [39]”

Line 294-295: Ok, but that does not mean that ecomorphological variation between close sciurid species is necessarily much lower than in other rodents (see also Cox et al., 2020 Journal of Zoology, on British red squirrels)

We are not implying any comparison in the variation of the tree squirrel species we studied and other rodent taxa, but rather that previous work has found tree squirrels to show conservatism through studies of phylogenetic signal [39], similarities among taxa in coronoid process shape [62], and lever arm ratios [63].

Line 309: Say "variation", you did not test for the disparity.

We implemented the change suggested by the reviewer.

Line 311: Same thing about use of disparity, and this result should be mentioned before.

We implemented the change suggested by the reviewer.

Line 312-313: I understand your point (plateau vs peak) but this is not highly explicit. It should be rephrased.

We have rephrased this sentence to make the point clearer regarding speculation towards a broad adaptive plateau being possible in this case. The modified sentence reads as: “Morphology can become more variable along an adaptive plateau rather than a sharp adaptive peak, contributing to such signals of greater variation [65,69].”

Line 313-315: This explanation might also be relevant in hybrids showing a high morphological variation and showing less differences with red squirrels...

We have added hybrids in contrast to Douglas squirrels for this explanation. The modified sentence now reads as: “The greater range of morphological variation in red squirrels and hybrids than in Douglas squirrels may reflect a wider adaptive plateau and a greater range of viable phenotypes for suture complexity and mandible shape.”

Line 320-321: I don't think the kind of food and its hardness in the hybrid zone has been mentioned before (e.g. in the introduction), while this information is crucial.

We have added a more detailed description of the forest composition and prevalent food items for the hybrid zone to the introduction to hopefully resolve this. The passage added to the introduction covering this topic reads as: “Within the hybrid zone, observational studies have shown that both parental squirrel species and their hybrids are syntopic and select similar food items in subalpine forests dominated by Engelmann spruce and subalpine fir [45,51], which are expected to be qualitatively soft cones.”

Line 322-326: Many information: make two sentences. Moreover, be careful of not overinterpreting your results in terms of roles of genetic processes, phenotypic plasticity and selection.

We have broken up and partially rephrased these two sentences to make them more clear. The modified sentences now read as: “The persistence of morphological differences in sympatry suggests divergence in the genetic architecture of some craniomandibular traits are maintained. This occurs despite potential homogenization effects of backcrossed hybridization, introgression of genetic variation, and perhaps some limitations in the role of phenotypic plasticity allowing convergent morphologies in a shared environment.”

Line 333-334: I would say "similar mandible shape". It is difficult to state that this is a convergent pattern in a context of hybridization, involving a part of the genetic background no longer shared in allopatric species. Moreover, the hypothesis of convergence is challenged by the differences observed between HZ populations regarding BFQ and cranial sutures.

We have rephrased this statement to “similar mandible shape” as recommended by the reviewer. We agree that this phrasing is more consistent with what we are able to claim about the observed patterns, given our results. The sentence now reads as: “These findings, however, contrast with our findings on mandibular morphospace, as both species display similar mandible shape”

Line 334-336: Ok, but this hypothesis does not explain the lower values observed for suture complexity (which might be also plastic contrary to incisor) in HZ Douglas squirrels compared with allopatric populations (cf hypothesis of weaker biomechanical demands for the different HZ populations).

While trending lower, the values for suture complexity were not significantly different between HZ Douglas squirrels and allopatric Douglas squirrels. For this hypothesis we are commenting on what has been learned regarding bite force in this study given statistically significant findings, and not necessarily the combination of suture complexity and bite force. 

Line 339-341: Ok, but wider range of cones properties than in the allopatric areas or not? Still unclear.

We have added framing for the cones found in the hybrid zone in the introduction, which reads as follows: “Within the hybrid zone, observational studies have shown that both parental squirrel species and their hybrids are syntopic and select similar food items in subalpine forests dominated by Engelmann spruce and subalpine fir [45,51], which are expected to be qualitatively soft cones.”. Absolute conditions of cone properties have not been quantitatively assessed; however, it is known that Engelmann spruce and subalpine fir are used by both species as a shared resource in the hybrid zone and are expected to be softer than the serotinous cones of lodgepole pine. 

Line 370-372: And cf comments on greater morphological variation in red squirrel and HZ populations (l.313-315).

We have added a detail about the greater morphological variation observed in hybrids compared to Douglas squirrel to this statement as well. The modified statement reads as: “One explanation for why hybrid morphologies resemble red squirrel morphologies more closely and have greater morphological variation than Douglas squirrel is the genetic dominance of red squirrel alleles underlying these morphological traits.”

Line 386: Hardnesses

We implemented the change suggested by the reviewer.

Line 398-400: Rephrase please, a part of the sentence is missing.

We have fixed this sentence to include the entirety of the intended point, which was that existing studies of such hybrids were largely laboratory-based and lacked full ecological context. The modified sentence reads as: “Despite the impressive breadth of research on craniomandibular traits in house mice hybrids, including developmental stability, genetic architecture, and gross morphology [24,84-86], existing studies are mostly laboratory-based and not necessarily true to ecology. In addition, the similar generalist diets of both mice subspecies can make it challenging to interpret the adaptive significance of morphological divergence.” 

Line 398: It should be defined for readers non-familiar with this terminology.

We have included a short definition for heterosis, in this case being an “enhanced hybrid phenotype”. 

Line 424-426: You should precise the number of specimens investigated for each analysis, especially if you excluded damaged specimens for some analyses (cf fig. 3 and 4).

We have added a statement to the methods that reads as “Some specimens had varying degrees of damage and were excluded from some analyses. Final sample numbers for analyses are shown in the respective figures for those analyses.”. 

Line 493: This is not relevant since you digitized all the dentary shape.

We modified this sentence to make the statement more relevant.

Line 509: Why did you also include the centroid size in these analyses, and why not only PC axes? It should have been interesting to perform these analyses without centroid size for accurate comparisons of conformations (at least in supp. data).

We included log10-transformed centroid size in this analysis, as it was found to be significantly different for the assessed populations and was expected to play an important role in the overlap between these taxa. Inclusion of size in this analysis is consistent with previous studies using this analysis to explore morphological overlap whereby centroid size differences were detected (Hedrick 2020, citation number 101).

---

## [Editor Report · Decision Letter 1]

23 Mar 2023

Craniodental divergence associated with bite force between hybridizing pine squirrels (Tamiasciurus)

PONE-D-22-29005R1

Dear Dr. Poorboy,

We’re pleased to inform you that your manuscript has been judged scientifically suitable for publication and will be formally accepted for publication once it meets all outstanding technical requirements.

Kind regards,

Cyril Charles

Academic Editor

PLOS ONE
---

## [Editor Report · Acceptance letter]

29 Mar 2023

PONE-D-22-29005R1 

Craniodental divergence associated with bite force between hybridizing pine squirrels (*Tamiasciurus*) 

Dear Dr. Poorboy:

I'm pleased to inform you that your manuscript has been deemed suitable for publication in PLOS ONE. Congratulations! Your manuscript is now with our production department. 

Kind regards, 

on behalf of

Dr. Cyril Charles 

Academic Editor

PLOS ONE